# IgE Downregulates PTEN through MicroRNA-21-5p and Stimulates Airway Smooth Muscle Cell Remodeling

**DOI:** 10.3390/ijms20040875

**Published:** 2019-02-18

**Authors:** Lei Fang, Xinggang Wang, Qingzhu Sun, Eleni Papakonstantinou, Chongteck S’ng, Michael Tamm, Daiana Stolz, Michael Roth

**Affiliations:** 1Pneumology & Pulmonary Cell Research, Departments of Internal Medicine & Biomedicine, University & University Hospital Basel, Petersgraben 4, CH-4031 Basel, Switzerland; lei.fang@unibas.ch (L.F.); sunqingzhu@nwafu.edu.cn (Q.S.); epap@med.auth.gr (E.P.); michael.tamm@usb.ch (M.T.); Daiana.Stolz@usb.ch (D.S.); 2Gynecological Endocrinology, Department of Biomedicine, University & University Hospital Basel, Hebelstrasse 20, CH-4031 Basel, Switzerland; xinggang.wang@unibas.ch; 3College of Animal Science and Technology, Northwest A&F University, Yangling 712100, China; 4Laboratory of Pharmacology, School of Medicine, Aristotle University of Thessaloniki, 54124 Thessaloniki, Greece; 5PersCellMed S’ng, CH-4450 Sissach, Switzerland; ctsng@hotmail.com

**Keywords:** airway smooth muscle cells, microRNA-21-5p, STAT3, mitochondria

## Abstract

The patho-mechanism leading to airway wall remodeling in allergic asthma is not well understood and remodeling is resistant to therapies. This study assessed the effect of immunoglobulin E (IgE) in the absence of allergens on human primary airway smooth muscle cell (ASMC) remodeling in vitro. ASMCs were obtained from five allergic asthma patients and five controls. Proliferation was determined by direct cell counts, mitochondrial activity by expression of cytochrome c, protein expression by immunoblotting and immuno-fluorescence, cell migration by microscopy imaging, and collagen deposition by cell based ELISA and RNA expression by real time PCR. Non-immune IgE activated two signaling pathways: (i) signal transducer and activator of transcription 3 (STAT3)→miR-21-5p→downregulating phosphatase and tensin homolog (PTEN) expression, and (ii) phosphatidylinositol 3-kinases (PI3K)→protein kinase B (Akt)→mammalian target of rapamycin (mTOR)→ribosomal protein S6 kinase beta-1 (p70s6k)→peroxisome proliferator-activated receptor gamma coactivator 1-α (PGC1-α)→peroxisome proliferator-activated receptor-γ (PPAR-γ)→cyclooxygenase-2 (COX-2)→mitochondrial activity, proliferation, migration, and extracellular matrix deposition. Reduced PTEN expression correlated with enhanced PI3K signaling, which upregulated ASMC remodeling. The inhibition of microRNA-21-5p increased PTEN and reduced mTOR signaling and remodeling. Mimics of microRNA-21-5p had opposing effects. IgE induced ASMC remodeling was significantly reduced by inhibition of mTOR or STAT3. In conclusion, non-immune IgE alone is sufficient for stimulated ASMC remodeling by upregulating microRNA-21-5p. Our findings suggest that the suppression of micoRNA-21-5p may present a therapeutic target to reduce airway wall remodeling.

## 1. Introduction

Asthma is a worldwide increasing health problem and approximately 60% of all patients suffer from allergic asthma [1]. Despite chronic inflammation, airway remodeling is a common pathology in asthma, which is currently regarded as a key toward understanding the pathogenesis and development of new therapies for asthma [2]. Deregulated mitochondria, specifically in airway smooth muscle cells (ASMC), have been shown to be a key driver of remodeling in asthma [3,4].

Airway wall remodeling in asthma is characterized by deranged epithelium, thickened sub-epithelial basement membrane, accumulated sub-epithelial extracellular matrix (ECM), and hyperplasia/ hypertrophy of ASMC [3]. Airway wall remodeling correlated with increased serum immunoglobulin E (IgE) levels in asthma patients and, in animal models, it was confirmed that IgE could induce remodeling [5,6]. Besides indirect activation of remodeling by IgE through mast cells, recent studies suggested that IgE directly activates ASMC, and, thereby, induces airway remodeling [7]. In vitro, ASMC from asthma patients constitutively expressed the high affinity immunoglobulin E receptor (FcεR-I), and the low affinity FcεR-II (CD23) receptors [8,9]. Thus, ASMC responded to non-immune IgE in the absence of allergens by increased proliferation and ECM production [8,9,10,11]. In human ASMC, IgE alone stimulated mitogen activated protein kinases (MAPK) as well as the signal transducer and activator of transcription (STAT3), which upregulates proliferation and remodeling [11]. As reviewed by Kim et al. [12], allergen bonded IgE activated the signaling pathway: phosphatidylinositol 3-kinases (PI3K)→ protein kinase B (Akt)→ mammalian target of rapamycin (mTOR) signaling. This signaling pathway is an important mediator of cell activity and tissue remodeling [13]. However, it is unclear if this molecular mechanism is involved in ASMC remodeling.

Altered mitochondria activity has been linked to the pathogenesis of various airway diseases including asthma, and presents a potential target for therapies for airway remodeling [4,14]. In ASMC, increased mitochondria correlated with hyper-constriction and intracellular calcium levels [12]. Furthermore, mitochondria were activated via PI3K and Akt in human ASMC [15]. 

Recently, the role of non-coding microRNAs contributing to airway wall remodeling in asthma has been described [16]. Deregulated microRNA-21-5p expression was considered as pro-fibrotic in chronic inflammatory lung diseases [17]. MicroRNA-21-5p increased ASMC proliferation and migration through down-regulation of the phosphatase and tensin homolog gene (PTEN), which is reduced in asthma [18]. Increased circulating microRNA-21-5p and microRNA-26a were suggested as a biomarker for allergies and wheezing in children [19]. In a clinical study, circulating microRNA-21-5p correlated with reduced Forced expiratory volume in 1 second (FEV1) in children with asthma [20]. Downregulation of microRNA-21-5p in pulmonary arterial smooth muscle cells induced apoptosis, while increased microRNA-21-5p prevented apoptosis [21]. In human aortic smooth muscle cells, mechanical stress upregulated microRNA-21-5p expression, which was linked to increased proliferation [22].

This investigation assessed the hypothesis that non-immune IgE in the absence of allergens directly affects ASMC remodeling by activating two independent signaling pathways, which enhance each other. We hypothesize that: IgE activates PI3K→Akt→mTOR, which enhances the synthesis of pro-inflammatory ECM components and increases mitochondria mass and activity. In parallel, IgE activates signal transducer and activator of transcription 3 (STAT3) and upregulates microRNA-21-5p. The latter inhibits phosphatase and tensin homolog (PTEN) expression, and, thereby, removes a potent inhibitor of Akt signaling, as shown in the Figure 1 (Graphic Figure abstract).

## 2. Results

### 2.1. IgE Stimulated ECM Deposition and Migration in ASMC from Asthmatic Patients

Western blot analysis revealed that FcεR-Iα, which is the alpha subunit of FcεR-I that recognizes extracellular IgE, as well as FcεR-II (CD23) were expressed by ASMC from asthma patients and controls (Figure 2A). Compared to ASMC of healthy controls, the quantitation of protein bands by image analysis showed significantly increased expression of FcεR-Iα in ASMC of asthma patients (*p* < 0.01), but not of FcεR-II (Figure 2A). The increased expression of FcεR-Iα in ASMC from asthmatic patients was confirmed by confocal microscopy (Figure 2B). 

Regarding the increased deposition of the extracellular matrix during airway wall remodeling, we confirmed the previously reported effect of non-immune IgE on the deposition of collagen type-I, and fibronectin by ASMC of asthma patients. IgE (1 μg/mL) significantly stimulated the deposition of collagen type-I and fibronectin by ASMC over 24 and 48 h, as determined by cell based ELISA (Figure 2C). IgE-induced collagen type-I deposition increased by 169.9 ± 20.3% at 24 h and by 188.9 ± 18.3% at 48 h compared to ASMC in the absence of IgE (Figure 2C, left panel). Compared to unstimulated ASMC, IgE-induced fibronectin deposition was increased by 176.3 ± 14.4% after 24 h and by 206.5 ± 18.4% after 48 h, as shown in Figure 2C. No difference was observed comparing IgE induced collagen and fibronectin deposition in ASMC obtained from asthma patients and controls.

The effect of IgE on ASMC migration was assessed in a model of wound repair, which was defined as a 2 mm scratch in a confluent ASMC layer (Figure 2D). The closure of the wounded area was monitored and measured by microscopy over 36 h. In the presence of IgE alone (1 μg/mL), ASMC migrated significantly faster into the wounded area compared to the absence of IgE. This effect became significant after 12 h (*p* < 0.01) when compared to unstimulated ASMC (Figure 2D). The effect of IgE on cell migration is depicted in more detail in Appendix A
Figure A1, as representative white balance pictures acquired by microscopy. No significant difference was observed comparing the effect of IgE on ASMC migration in cells from asthma patients and controls. The fast closure of the wounded area is mainly due to migration than proliferation. The latter effect would need more than 36 h to achieve significance. Single cell movement was monitored by a single investigator in a specific area of the wound.

### 2.2. IgE Upregulated the Expression of Mitochondria-Related Genes and Proteins in ASMC

The effect of IgE on mitochondria-regulating key meditators, including cytochrome c Oxidase Subunit 2 (COX-2), Peroxisome Proliferator-Activated Receptor-γ (PPAR-γ), and Peroxisome Proliferator-Activated Receptor γ Coactivator-1α (PGC-1α) in ASMC was determined on the pre-transcriptional and post-transcriptional level in ASMC obtained from asthma patients and controls. 

Regardless of the cell donor’s diagnosis (asthma, control), IgE stimulated COX-2 mRNA expression, which increased significantly after 3 h (*p* < 0.05) and reached a 4.5-fold increase (*p* < 0.01) after 24 h, as compared to unstimulated cells (Figure 3A). Additionally, independent from the diagnosis, IgE significantly upregulated the expression of PGC-1α mRNA after 6 h (*p* < 0.05) and this effect remained significant up to 48 h (*p* < 0.001) with a 20-fold increase (Figure 3A). The gene expression of the mitochondria activator PPAR-γ increased by two to three fold after 24 h (*p* < 0.05) and continued to increase until 48 h (*p* < 0.05) after IgE stimulation (Figure 3A). 

On the protein level, IgE-induced COX-2 protein expression significantly after 6 h (*p* < 0.05), and remained at a high level up to 48 h (*p* < 0.05), as shown by a representative Western blot from ASMC of an asthma patient (Figure 3B), and subsequent analysis of protein band optical density by the imageJ program. Comparing the data between asthma and non-asthma derived cells, we observed no significant disease effect. The data of total and phosphorylated proteins are presented as bar charts in Figure 3B. Two hours after IgE treatment, a significant upregulation of PGC-1α protein level was determined (*p* < 0.05), which peaked at 6 h (*p* < 0.01) and reduced thereafter (Figure 3B). IgE significantly increased the expression of total PPAR-γ protein after 3 h (*p* < 0.05), which steadily increased over 48 h (Figure 3B). The stimulatory effect of IgE on PPAR-γ phosphorylation was significant after 15 min (*p* < 0.05), and further increased over the next 48 h (Figure 3B) in both ASMC of asthma patients and controls. 

The stimulatory effect of IgE on ASMC mitochondria activity was determined by cytochrome c expression. Confocal microscopy showed a fast IgE-induced increase of cytochrome c in the cytosol, which became significant at 6 h and remained significant up to 48 h (Figure 3C). Cytochrome c expression was quantified by image analysis in IgE stimulated and unstimulated ASMC. The mechanism was similar in ASMC obtained from asthma patients and controls. The results are depicted as bar charts beside the immunofluorescence microscopic photographs (Figure 3C).

### 2.3. IgE Acted via PI3K→Akt→p70s6k and STAT3→miR-21→PTEN Signaling Pathways

The kinetic of IgE-induced PI3K→Akt→phosphorylation of ribosomal protein S6 kinase beta-1 (p70s6k) was determined by Western blotting and the optical density of the protein bands was determined by imageJ in five blots. A representative Western blot and the kinetic of the proteins as the line chart is shown in Figure 4A. Compared to unstimulated ASMC, IgE significantly increased the phosphorylation of PI3K within 15 min (*p* < 0.05), peaking after 1 h, and declining to baseline levels thereafter (Figure 4A). This effect of IgE was made parallel to increase phosphorylation of Akt, which became significant at 15 min (*p* < 0.05) and further increased over 48 h (Figure 4A). Similarly, IgE induced the phosphorylation of p70s6k, which became significant at 30 min (*p* < 0.05) and the protein remained phosphorylated until 48 h (Figure 4A).

Immunofluorescence microscopy confirmed the IgE-induced phosphorylation of PI3K in sub-membranous protrusion sites, intracellular areas, and in the nuclei, as shown for 30 min and 1 h in Figure 4B. This effect of IgE was followed by mTOR phosphorylation and accumulation of the protein in the nuclei, which became significant from unstimulated ASMC after 24 h and further increased up to 48 h (Figure 4B). No difference of the mechanism was detected comparing ASMC from asthma patients to controls.

IgE upregulated the phosphorylation of Akt and p70s6k in ASMC derived from asthma patients and controls. IgE stimulated the phosphorylation of STAT3 within five min, which became significant at 30 min (*p* < 0.05) and was maintained over 48 h (Figure 4C). In parallel, the expression of PTEN significantly decreased 30 min (*p* < 0.05) after IgE stimulation, as shown by the representative Western blot and subsequent image analysis (Figure 4C). Assessed by immunofluorescence microscopy, PTEN expression was detectable in unstimulated ASMC in the cytosol and the nuclei. IgE, significantly, downregulated the expression of PTEN at 24 and 48 h (Figure 4D). No significant differences of this mechanism were observed comparing results from asthmatic and non-asthmatic ASMC.

The expression of miR-21-5p, as determined by quantitative real time-PCR, was significantly higher in ASMC from asthma patients (*n* = 5, *p* < 0.01) as compared to ASMC of non-asthma controls (*n* = 5) (Figure 4E). Stimulation by IgE induced microRNA-21-5p expression by ASMC of controls and further increased the expression in ASMC of asthma patients. This effect was time dependent, as shown in Figure 4F. The data suggest that the IgE induced expression of microRNA-21-5p peaked after 24 h (*p* < 0.01) and declined thereafter (Figure 4F).

In Figure 5A, we provide evidence based on Western blots that the inhibition of mTOR by rapamycin significantly reduced the phosphorylation of p70s6k, while the expression of total p70s6k was not affected. This effect occurred in both IgE stimulated and unstimulated ASMC, and was independent of asthma. A similar reducing effect on IgE induced phosphorylation was observed after the inhibition of PI3K by LY294002 (Figure 5A). In contrast, the inhibition of STAT3 by S31-201 only partially reduced the phosphorylation of p70s6k. However, the effect was still significant (*p* < 0.05, Figure 5A) in both ASMC of asthma patients and controls.

In order to determine the contribution of the signaling cascade described above in ASMC controlled airway wall remodeling, the deposition of collagen type-I and fibronectin was determined in the presence and absence of IgE and the signaling inhibitors, as shown in Figure 5B. IgE induced deposition of collagen type-I, which was completely prevented in the presence of rapamycin, LY294002, or S31-201 (Figure 5B). IgE-induced fibronectin deposition was reduced to baseline value by rapamycin and LY294002, while the STAT3 inhibitor reduced the fibronectin deposition incompletely (Figure 5B). The effects of the signaling inhibitors on collagen and fibronectin deposition were similar in ASMC of asthma patients and controls. 

IgE-induced migration of ASMC was inhibited by all three signal inhibitors, as shown in Figure 5C. No significant inhibitory effect of signaling inhibition was observed in the absence of IgE (Figure 5C). More details of IgE induced ASMC migration and its inhibition is provided in Appendix A. No significant difference of the effect was observed comparing ASMC derived from asthma patients or controls. Furthermore, the reduced wound closing cannot be explained by reduced proliferation because the observation period was too short.

Next, we assessed the effect of the signaling inhibitors on IgE-induced mitochondria activity by immunofluorescence microscopy (Figure 5D). As depicted in the image analysis panel (bar chart Figure 5D), the inhibitors of either PI3K, mTOR, or STAT3 were similarly effective and significantly reduced Cytochrome c expression in IgE-treated ASMC (Figure 5D). Cytochrome c expression was quantified image analysis of 10 single cells in five individual ASMC lines, totaling 50 single cells for each condition shown in Figure 5D, and the data is displayed as a bar chart.

### 2.4. MicroRNA-21-5p is Essential for IgE-Induced ASMC Remodeling

Regardless of the diagnosis of the cell donors and in the absence of IgE, the overexpression of microRNA-21-5p in ASMC increased collagen type-I and fibronectin deposition, while anti-sense microRNA-21-5p reduced both (Figure 6A). Similarly, the overexpression of microRNA-21-5p, in the absence of IgE, significantly increased ASMC migration, while microRNA-21-5p inhibition delayed ASMC migration (Figure 6B). 

The overexpression of microRNA-21-5p in ASMC decreased baseline PTEN expression in a dose-dependent manner (Figure 6C). In contrast, overexpression of microRNA-21-5p increased p70s6k phosphorylation in a concentration (Figure 6C) and time-dependent manner (Figure 6D). Inhibition of microRNA-21-5p increased PTEN expression and inhibited p70s6k phosphorylation, which was concentration dependent (Figure 6C) and time dependent (Figure 6D). 

Mitochondrial activity was monitored by cytochrome c expression, which was significantly upregulated in ASMC after overexpression of microRNA-21-5p (Figure 6E). In contrast, the inhibition of microRNA-21-5p downregulated the expression of cytochrome c (Figure 6E). The expression of cytochrome c was quantified by image analysis with a total of 50 single cells and the data is displayed as bar charts in Figure 6E. All effects of the microRNA-21-5p modulation was similar in ASMC of asthma patients and controls.

## 3. Discussion

The ASMC driven airway wall remodeling in allergic asthma is a major irreversible pathology that was assumed to be the result of chronic inflammation [23]. However, other studies provided evidence suggesting that this pathology is independent of inflammation or disease duration [24,25]. The present study assessed the hypothesis that non-immune IgE is sufficient to induce ASMC remodeling by direct activation of two independent but interacting signaling pathways: (i) STAT3→microRNA-21-5p induced downregulation of PTEN and (ii) activation of Akt→mTOR→p70s6k, as summarized in the graphical abstract. Thus, the IgE-induced ASMC remodeling is mediated by upregulated microRNA-21-5p and subsequent support of mTOR signaling.

Increased serum IgE levels is one of the characteristic pathology of allergic asthma, which persists for several weeks after the clearance of allergens from the blood [5,6,7,26,27,28]. However, aside from initiating inflammation after allergen crosslinking, there is little evidence in clinical studies that IgE directly contributed to airway wall remodeling. Studies investigating the effect of anti-IgE therapy in allergic asthma reported symptom reduction, lung function improvement, and reduced immune cell infiltration, while reduced airway wall thickness was not observed [7,26]. 

Earlier studies showed that ASMC express the high affinity IgE-receptor FcεR-I, and responded to extracellular IgE by increased proliferation and ECM deposition [8,9,11]. Thus, this implies that the presence of non-immune IgE in the presence of allergens may be sufficient to activate ASMC. Furthermore, increased mitochondrial activity in ASMC of asthma patients was reported and causative for proliferation [29]. In this study, confocal microscopy and protein electrophoresis confirmed a higher expression of FcεR-I in ASMC of asthma patients as compared to cells of non-diseased controls [9,11]. The activation of ASMC by IgE increased the expression of mitochondrial proteins such as COX-2 and cytochrome c. 

In a mouse model, prolonged antigen exposure induced airway wall remodeling, but did not increase inflammation further [30]. In humans, treatment with neutralizing anti-IgE antibody (Omalizumab) reduced the thickness of the bronchial basement membrane by inhibiting eosinophil’s activity [31]. In a second study, the anti-IgE antibodies slowed the progress of remodeling, which was attributed to reduced inflammation, but not to a direct inhibition of ASMC remodeling [32]. Furthermore, the level of circulating IgE in patients with allergy was not immediately reduced by neutralizing IgE antibodies [33,34]. Thus, the direct inhibitory effect of Omalizumab on airway remodeling remains to be proven. Irreversible ASMC and myo-fibroblast hypertrophy and hyperplasia were described in childhood asthma without inflammation [35]. In volunteering patients with light asthma, an experimental challenge with dust mite allergens or carbachol-caused airway wall remodeling within four days without inducing inflammation [36]. Therefore, inflammation should no longer be regarded as the only cause of airway wall remodeling in allergic asthma.

It has been shown that activated PPAR-γ regulated glucose/lipid metabolism as well as the inflammation response and may act through NF-κB, p38-MAPK, or Erk1/2-MAPK signaling [37]. PGC-1α was a co-activator for many transcription factors that regulated nucleic-genes, mitochondrial-genes, and mitochondrial biogenesis [38]. PGC-1α was also described as the core factor controlling mitochondrial function in skeletal muscle cell plasticity [39]. In this study, both PPAR-γ and PGC-1α was significantly upregulated after IgE stimulation, and a fast phosphorylation of PPAR-γ. Based on these results, we hypothesized that IgE activates PGC-1α and PPAR-γ through Akt→mTOR signaling, which leads to airway wall remodeling by ASMC. 

Abnormal mitochondrial function in asthmatic ASMC has been previously reported [4,14,29]. Besides providing energy for ASMC remodeling, the increased mitochondria function may also upregulate the production of pro-inflammatory reactive oxygen species (ROS) [40]. In this study, increased COX-2 and Cytochrome c expression by ASMC after IgE stimulation supported the hypothesis that IgE promoted ASMC remodeling by deregulating mitochondrial activity.

Activated mTOR signaling participated in IgE-activated mast cells and lymphocytes in asthma, while no such data was reported for ASMC remodeling [13]. Akt → mTOR signaling controlled the synthesis of collagen type-I in hepatic stellate cells [41], vascular smooth muscle cells [42], and tubular cells [43]. It was, therefore, regarded as an essential signaling pathway in remodeling. In this study, PI3K phosphorylation occurred 5 min after IgE stimulation, which was followed by Akt and p70s6k activation. Inhibition of this signaling pathway prevented IgE induced mitochondria activity, ASMC migration, and ECM deposition, which indicates its essential role for remodeling. The persistent activation of Akt→p70s6k in IgE-treated ASMC suggested a second molecular signaling pathway that prolonged the activation of mTOR signaling through STAT3 phosphorylation, which was followed by upregulated microRNA-21-5p and PTEN knockdown. 

Circulating microRNA-21-5p was suggested as a biomarker for allergic inflammatory disease during childhood [44]. Increased expression of microRNA-21-5p on recurrent wheezing in atopic children was reported [19]. In steroid resistant asthma, microRNA-21-5p suppressed histone deacetylase-2 through PI3K [45]. In line with these studies, we observed that microRNA-21-5p expression was significantly higher in ASMC of asthma patients as compared to cells of non-asthma controls. In isolated human ASMC, microRNA-21-5p was significantly upregulated by IgE both in asthma and non-asthma ASMC. With regard to the regulation of miR-21 by IgE, STAT3 is an essential transcription factor for miR-21 expression [11]. Activation of mitochondrial STAT3 played a major role for IgE-induced mast cell exocytosis [46]. In epidermal mast cells, STAT3 contributed to FcεR-I activated phosphoglandin-2 and VEGF synthesis [47]. Earlier investigation has shown that microRNA-21-5p targeted the Akt inhibitor PTEN and, thereby, modulate ASMC proliferation [18]. In this context, the loss of PTEN contributed to ASMC remodeling. In line with these studies, we observed that microRNA-21-5p reduced PTEN expression in IgE-treated ASMC and, thereby, supported pro-remodeling signaling of mTOR.

The results of this study showed that microRNA-21-5p is essential for IgE-dependent ASMC remodeling and, thus, presents a target for therapies of airway wall remodeling in asthma. These findings are in line with earlier studies, which suggests that microRNA-21-5p plays a role in asthma [19,20,40,41]. In order to reduce airway wall remodeling, future studies have to develop therapeutic strategies to control the expression of microRNA-21-5p with a cell-type specific application.

## 4. Materials and Methods

### 4.1. Tissue Biopsies and Primary Cell Cultures

Human bronchial biopsies were obtained from patients with asthma or from non-asthma/non-COPD (controls) that underwent bronchoscopy for diagnostic reasons at the Clinic of Pneumology, University Hospital Basel, Switzerland. Each patient gave a written consent for using one additional biopsy sample for scientific investigations. The local Ethics Board (EKNZ 2016-01057, 7 July 2016, Ethic commission Northwest- and Central Switzerland, EKNZ) approved the protocol. The clinical parameters of all tissue donors are presented in Table 1.

The primary ASMCs were isolated from the biopsies (obtained from five patients with asthma and five controls) after removing the epithelium and cultivated in a specific medium (DMEM supplemented with 5% fetal calf serum, 1mM sodium pyruvate, 1 × MEM vitamin mix, 1 × non-essential amino acid mix, and 10mM HEPES; all Thermo Fisher Scientific, Basel, Switzerland) [48]. Positive staining for fibrillar α-smooth muscle actin (ab5694, Abcam, Cambridge, UK) and negative staining for fibrillar fibronectin (#sc-9068, Santa Cruz, Santa Cruz, CA, USA) and E-cadherin (#610181, BD Bioscience, Allschwil, Switzerland) characterized ASMCs. All experiments were performed between passages 3 and 8.

### 4.2. ASMC Treatments 

ASMC from patients with asthma were seeded into 6-well plates and allowed adherence for 24 h, which was followed by overnight serum deprivation in DMEM containing 0.1% serum (control medium). ASMCs were further incubated in the absence (control) or in the presence of human non-immune IgE (human IgE, 1.0 μg/mL, #DIA-HE1, BioPorto Diagnostic, Hellerup, Denmark) for up to 48 h.

Cell signaling pathways were confirmed by selective chemical inhibitors, which were added to ASMC 1 h before IgE stimulation: (a) STAT3 inhibitor at 20 μM (#S3I-201, Sigma-Aldrich, Basel, Switzerland), (b) PI3K inhibitor at 1 μM (#LY294002, Sigma-Aldrich), and c) mTOR inhibitor Rapamycin at 10 nM (#R8781, Sigma-Aldrich). 

For microRNA functional evaluation, ASMCs were seeded into 6-well plates and transiently transfected using HiPerfect Reagent Kit (#301705, Qiagen, Hombrechtikon, Switzerland). Synthetic miR-21-5p mimic and inhibitor were from Qiagen (#MSY0000076, #MIN0000076).

### 4.3. Real-Time Quantitative PCR (RT-qPCR)

Total RNA was purified from ASMC by using the Quick-RNA Mini Prep kit (#R1055, Zymo Research, Freiburg, Germany) and was followed with cDNA synthesis by High-Capacity cDNA Reverse Transcription Kit (#4368814, Thermo Fisher Scientific). To detect miR-21-5p, the miRNA-specific stem loop reverse transcription (RT) reaction was used to synthesize cDNA by Mir-XTM miRNA First-Strand Synthesis kit (#638315, Clontech Takara, Göteborg, Sweden). Real-time quantitative PCR (RT-qPCR) was performed by the Applied Biosystems 7500 Real-time system with FastStart Universial SYBR Green Master Rox (#04913850001, Roche Diagnostic, Rotkreuz, Switzerland) for quantification. The mRNA expression levels were normalized to GAPDH, while miR-21-5p expression was normalized to U6 snRNA. Purity of PCR products was confirmed by melting curve analysis, and the final data was analyzed using the 2^−ΔΔ*C*t^ algorithms. 

### 4.4. Primer Sequence Details 

Peroxisome Proliferator-Activated Receptor Gamma Coactivator-1α (PGC-1α, forward: 5′-CCT GCA TGA GTG TGT GCT CT-3′ and reverse: 5′-CAG CAC ACT CGA TGT CAC TCC-3′); Peroxisome Proliferator-Activated Receptor-gamma (PPAR-γ, forward: 5′-CCG TGG CCG CAG AAA TG-3′ and reverse: 5′-AGG AGT GGG AGT GGT CTT CC-3′); and Cytochrome c Oxidase Subunit 2 (COX-2, forward: 5′-GTG CAA CAC TTG AGT GAA TGA T-3′ and reverse: 5′-AGC AAT TTG CCT GGC CCA CT-3′). Primer sequence for miR-21-5p was: 5′-TAG CTT ATC AGA CTG ATG TTG A-3′. All primers were synthesized and purchased from Microsynth, Basel, Switzerland). The conditions for real-time PCR was: 95 °C and 30 s for template denature, 60 °C as the annealing and extension temperature for mRNA, and 55 °C for microRNA with 40 cycles to complete.

### 4.5. Western Blot

ASMCs were washed once with cold DPBS (#14190, Thermo Fisher Scientific) and lysed by RIPA buffer (#R0278, Sigma-Aldrich). Cell lysates were centrifuged at 12,000 rpm (10 min) and the protein concentration of the supernatant was measured by using the BCA protein assay kit (#23227, Thermo Fisher Scientific). Equal amounts of denatured proteins (20 μg) were separated in 4%–12% SDS–PAGE (#M41212, GenScript, Piscataway, NJ, USA), and, subsequently, transferred onto a nitrocellulose membrane (#88018, Themo Fisher Scientific). Proteins of interest were detected by specific antibodies, including total- (t-)PI3K (#4292), and phosphorylated- (p-)PI3K (#GTX132597), t-AKT (#4691) and p-AKT (#4060), t-p70S6 kinase (#2708) and p-p70S6 kinase (#9205), t-STAT3 (#9139) and p-STAT3 (#9145), PTEN (#9188, which were all purchased from Cell Signaling Technology, Cambridge, UK), PGC-1α (#ab54481, Abcam), t-PPARγ (#sc-7273, Santa Cruz), and p-PPARγ (#sc-28001, Santa Cruz). GAPDH (#2118, Cell Signaling Technology) was used as an endogenous control for a semi-quantification measure. Protein bands were visualized by Luminata Forte Western HRP substrate (#WBLUF, Sigma-Aldrich) after binding specie-specific secondary HRP conjugated antibodies (#7076 and #7074, Cell Signaling Technology).

### 4.6. Immunofluorescence 

For immunofluorescence analysis, ASMC cells were seeded on 10 mm sterile glass slides. Following experimental protocols, cells were fixed (2 × 5 min) by 4% paraformaldehyde (in DPBS). Immuno-fluorescence staining was performed by p-mTOR (#5536, Cell Signaling Technology), p-PI3K (#GTX132597, GeneTex, Irvine, CA, USA), and PTEN (#9188, Cell Signaling Technology). Antibodies were diluted, according to the manufacturer’s instructions in PBS containing 2% bovine serum albumin and incubated at 4 °C overnight, followed by three washes and subsequent staining with Alexa488 goat anti-rabbit IgG (#11008, Thermo Fisher Scientific). Cytoskeleton counter staining was performed by TRIC-Phalloidin (#P1951, Sigma-Aldrich), while nuclei were stained by DAPI (#D1306, Thermo Fisher Scientific). Images were acquired by the OLYMPUS BX63 upright fluorescence microscope under the 20x objective. The combined 3-channel images in the figures are generated by FIJI.

### 4.7. Confocal Microscopy

ASMCs were seeded onto coverslips in 24-well plates and incubated under various conditions for up to 48 h. Afterwards, ASMC were fixed by 4% formaldehyde (15 min, room temperature), washed once with DPBS, permeabilized by 0.15% Triton (15 min., room temperature), and blocked with 5% bovine serum albumin (BSA) in DPBS for 1 h. The primary mouse anti-Cytochrome c (#556432, Thermo Fisher Scientific) antibody or FITC anti-human FcεR-Iα (#334607, BioLegend, San Diego, CA, USA) was stained overnight at 4 °C. Alexa488-labeled anti-mouse secondary antibody (#11001, Thermo Fisher Scientific) was used to visualize Cytochrome c. F-actin is stained with TRIC-Phalloidin. Nuclei were stained with DAPI. Coverslips were mounted with Vectashield Antifade Mounting Medium (H-1000, Vector Laboratories, Burlingame, CA, USA). Images were acquired by using the Nikon Confocal A1 microscope with a 40× 1.3NA FI oil objective of 0.225 μm Z-stack step. The printed images are presented as Z-stack projection generated by imaging software FIJI. Single cell (50 cells/group) cytochrome c expression was calculated by FIJI with the “3D objects counter” plugin and finally normalized to the control group after 6 h. The cellular FcεR-Iα expression level was measured by the “Analyze Particle” feature of FIJI, quantified by an average of 10 cells/cell line, five cell lines/group, and analyzed by one-way ANOVA.

### 4.8. Collagen Type-I and Fibronectin ELISA 

Collagen-I and fibronectin deposition was determined by cell-based ELISA, as described previously [49]. ASMC (1 × 10^4^ cells/well) were seeded into 96-well plates and then treated with different conditions before fixation in 4% formaldehyde (15 min). Antigen-Retrieval was induced by the citric acid method (30 min, 37 °C) and un-specific binding was blocked by 3% BSA (1 h). Collagen type-I deposition was detected by anti-collagen-1A1 (#ab34710, Abcam, diluted 1:5000) and fibronectin was detected by anti-fibronectin antibody (#ab23750, Abcam, diluted 1:5000). After overnight incubation (4 °C), cells were washed three times with T-PBS (0.05% Tween20 in PBS), and incubated for 1 h with a secondary HRP-conjugated antibody (#7076, Cell Signaling Technology). After another three washes with PBS, TMB substrate (#T0440, Sigma-Aldrich) was added (room temperature, 20 min). The reaction was terminated by adding 1M HCl and the absorption was determined at 450 nm (Synergy H1 Hybrid Microplate Reader, BioTek, Basel, Switzerland). All experiments were performed in quadruplicate.

### 4.9. Cell Migration Assay

Cell migration was recorded by using time serial photography (OLYMPUS IX50 system) after mechanical wounding on a confluent cell layer, under selective designed experimental culture conditions at 0, 12, 24, and 36 h. The cell migration comparison was achieved by measuring the wound closure rate at a time point of 24 h. 

### 4.10. Statistical Analysis

The primary null-hypothesis was that IgE had no effect on any parameter of ASMC remodeling. A second null-hypothesis is that neither signaling inhibition nor microRNA modification modified IgE induced ASMC remodeling. All data are expressed as mean ± SEM, and were analyzed by unpaired one-way ANOVA and a subsequent Student’s *t*-test. *p*-values < 0.05 were considered as statistically significant. Statistical analysis was performed by software GraphPad-Prism7.

## 5. Conclusions

Non-immune IgE stimulated ASMC remodeling by upregulating microRNA-21-5p, which, thereby, downregulated PTEN and, thus, supported mTOR signaling. The data suggests that microRNA-21-5p is a potential target for novel therapies that aim to reduce airway wall remodeling in asthma.

## Figures and Tables

**Figure 1 ijms-20-00875-f001:**
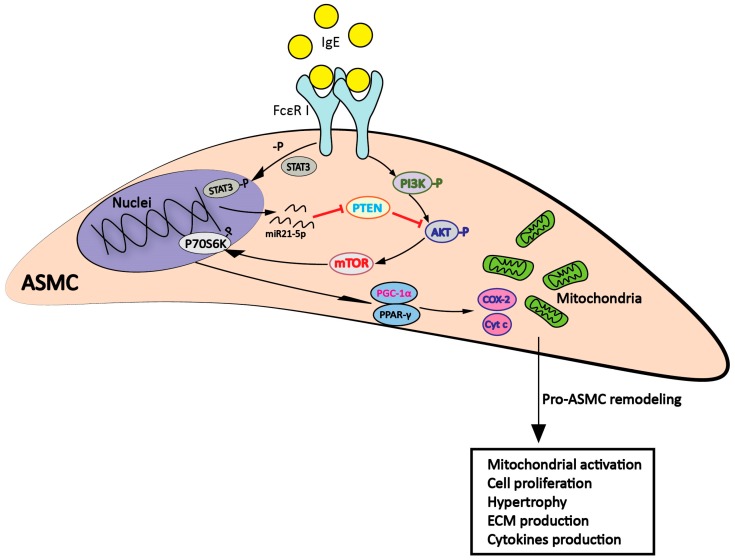
Graphic abstract demonstrated that non-immune IgE activated two signaling pathways in ASMC cells: (i) STAT3→miR-21-5p→downregulating PTEN expression, and (ii) PI3K→Akt→mTOR→p70s6k→PGC1-α→PPAR-γ→COX-2→mitochondrial activity, proliferation, migration, and extracellular matrix deposition. Reduced PTEN expression correlated with enhanced PI3K signaling, which upregulated ASMC remodeling. (ASMC: airway smooth muscle cell; STAT3: signal transducer and activator of transcription 3; PTEN: phosphatase and tensin homolog; PI3K: phosphatidylinositol 3-kinases; AKT: protein kinase B; mTOR: mammalian target of rapamycin; p70s6k: ribosomal protein S6 kinase beta-1: PGC1-α: peroxisome proliferator-activated receptor gamma coactivator 1-α; PPAR-γ: peroxisome proliferator-activated receptor-γ; COX-2: cyclooxygenase-2).

**Figure 2 ijms-20-00875-f002:**
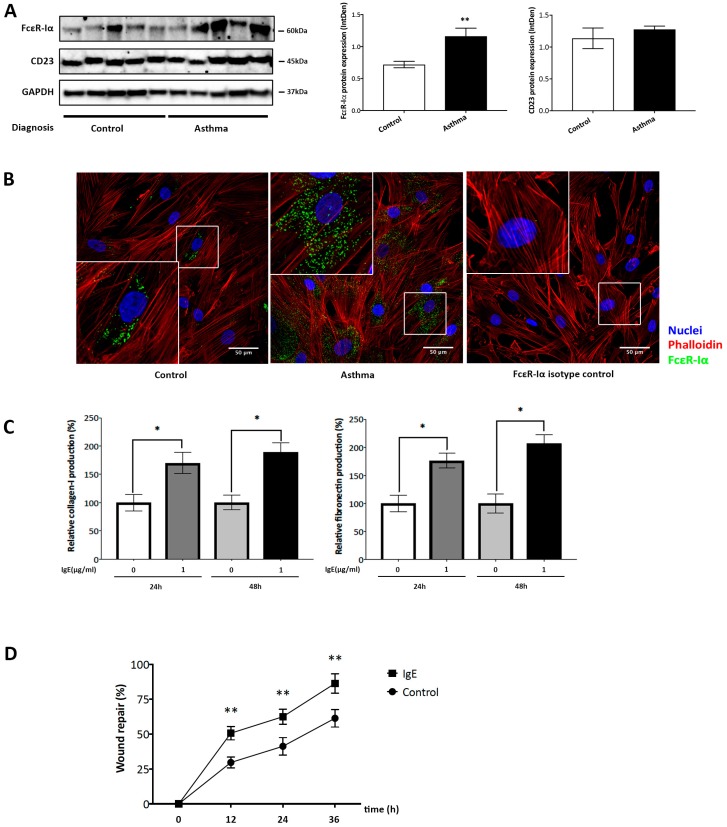
IgE receptor expression, IgE stimulated ECM deposition, and ASMC migration. (**A**) Western blot analysis of FcεR-Iα and FcεR-II expression in ASMC from non-asthma controls (*n* = 5) and asthma patients (*n* = 5). Protein quantitation was performed by Image J software. Bars represent mean ± SEM. ** *p* < 0.01. (**B**) Representative confocal microscopy images of FcεR-Iα and FcεR-II expression by ASMC of non-asthma and asthma patients: FcεR-Iα-FITC (green). TRIC-Phalloidin (red) for F-actin and DAPI (blue) for nuclei. (600X magnification in enlarged boxes) Similar results were obtained in all other cell lines. (**C**) Cell-based ELISA assessed IgE-induced deposition of collagen type-I and fibronectin by asthmatic ASMC at 24 and 48 h. Bars represent mean ± SEM of quadruplicated measurements performed in ASMC of asthma patient (*n* = 5), * *p* < 0.05. (**D**) Cell migration was assessed by measuring the width of a wound at 12, 24, and 36 h in the absence (control) or presence of IgE. Data points represent mean ± SEM from five independent experiments performed in cells obtained from five asthma patients. ** *p* < 0.01. Detailed images are presented in Appendix A
Figure A1.

**Figure 3 ijms-20-00875-f003:**
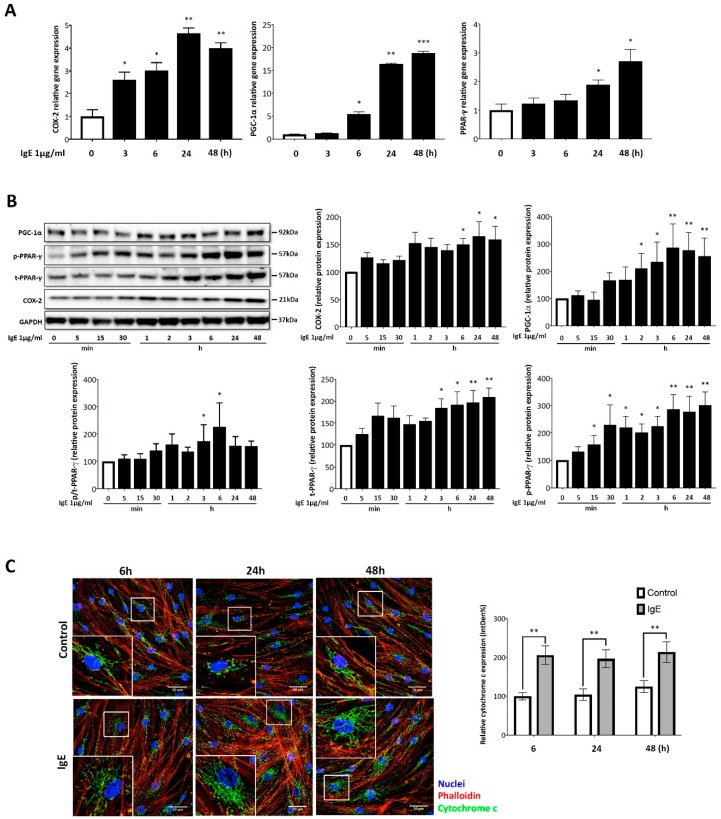
IgE stimulated mitochondria function-regulating genes and proteins. (**A**) IgE-induced gene expression of COX-2, PGC-1α, and PPAR-γ assessed by real-time PCR in ASMC of asthma patients (*n* = 5). Data are expressed as the percentage of none-treated cells at the same time point and bars represent mean ± SEM at the respective time points. * *p* < 0.05, ** *p* < 0.01, *** *p* < 0.001. (**B**) Western blot analysis for PGC-1α, COX-2, total- and phosphorylated-PPAR-γ in asthma patient’s ASMC (*n* = 5). Quantitation of the protein bands was performed with the Image J software. Bars represent mean ± SEM as the percentage of control at the respective time points. * *p* < 0.05, ** *p* < 0.01. (**C**) Representative confocal microscopy images for cytochrome c (green), F-actin (TRIC-Phalloidin, red), and nuclei (DAPI, blue) (600X magnification in enlarged boxes). Quantitation of fluorescence for cytochrome c was performed by Image J software as integrated density percentage of 10 single cells of asthma ASMC (*n* = 5). Bars represent mean ± SEM results obtained from 50 cells. ** *p* < 0.01, ASMC in the absence of IgE at 6 h were defined as 100%.

**Figure 4 ijms-20-00875-f004:**
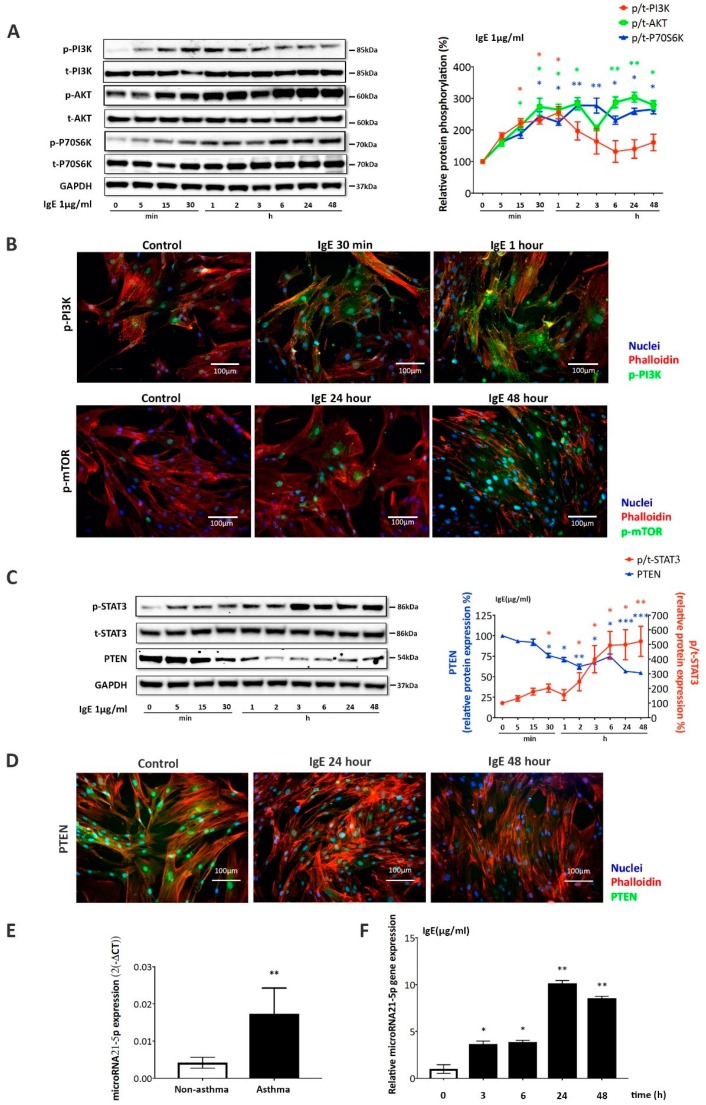
IgE activated PI3K→ AKT → mTOR and STAT3 → microRNA-21-5p, down-regulating PTEN in ASMC. (**A**) Representative Western blot kinetic of IgE-induced PI3K, AKT, and P70S6K phosphorylation in ASMC, and quantitative analysis of all five blots by Image J (line chart). Data points represent mean ± SEM experiments performed in ASMC of five asthma patients. * *p* < 0.05, ** *p* < 0.01. (**B**) Representative immunofluorescence microscopy of IgE (1 μg/mL) induced PI3K phosphorylation (upper panel, green) at 30 min and 1 h, and mTOR phosphorylation (lower panel, green-turquois) accumulation in the nuclei at 24 and 48 h. (nuclei: blue, F-actin: red). (**C**) Representative Western blots (right panel) and quantitation analysis by image J (left panel) of STAT3 phosphorylation and PTEN expression in response to IgE (1 μg/mL). Data points represent mean ± SEM of repeated measurements in ASMC of five asthma patients. * *p* < 0.05, ** *p* < 0.01, *** *p* < 0.001. (**D**) Representative immunofluorescence images of PTEN (green-turquois) expression in the presence of IgE. F-actin is indicated by TRIC-Phalloidin (red) and nuclei by DAPI (blue). (**E**) Real-time PCR analysis of microRNA-21-5p expression in ASMC of five asthma patients and non-asthma controls (*n* = 5). * *p* < 0.05. (**F**) Kinetic of microRNA-21-5p expression by Real-time PCR. Bars show mean ± SEM of experiments performed in ASMC of five asthma patients. MicroRNA-21-5p expression at time 0 was set as 1. * *p* < 0.05. ** *p* < 0.01.

**Figure 5 ijms-20-00875-f005:**
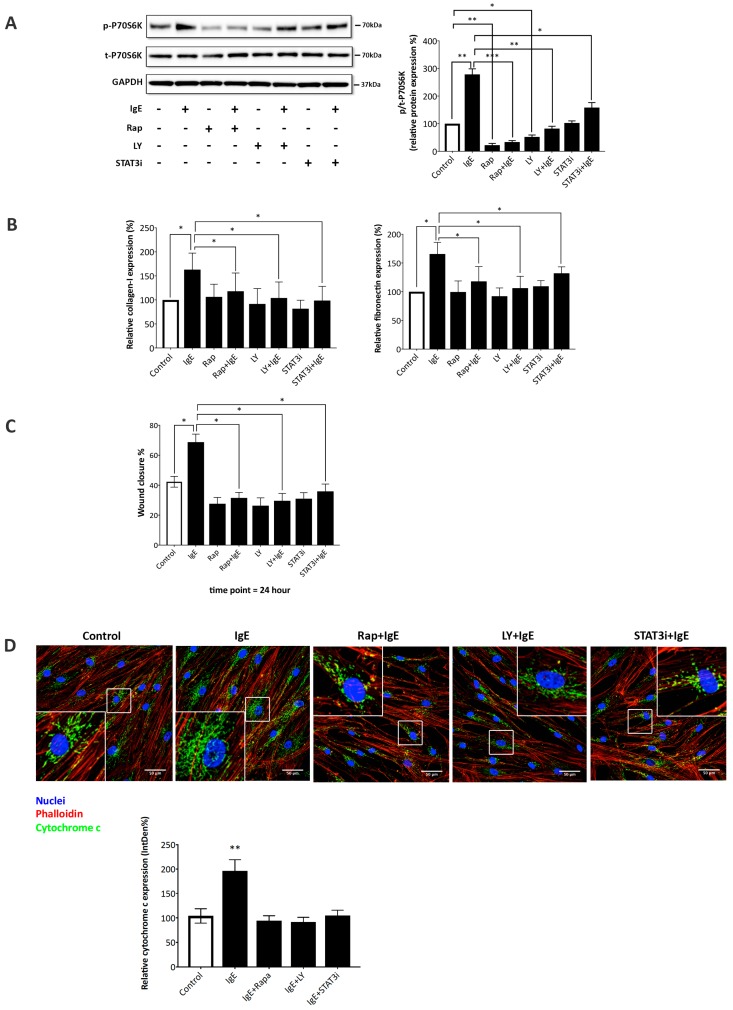
Signaling inhibitors blocked IgE-induced P70S6K and mitochondria activation. (**A**) Representative Western blot of the kinetic of IgE-induced phosphorylation of P70S6K, which was blocked by signaling inhibitors for rapamycin (Rap), LY294002 (LY), and S3I-201 (STAT3i). Left panel shows the quantitative analysis Western blots by Image J. Bars represent mean ± SEM P70S6K expression form ASMC of five asthma patients * *p* < 0.05, ** *p* < 0.01, *** *p* < 0.001. (**B**) Signal inhibitors reduce collagen-I and fibronectin deposition by ASMC after 24 h, as determined by ELISA. Bars represent mean ± SEM of quadruplicate measurements in ASMC five asthma patients. The “control” depicts deposition in the absence of IgE. * *p* < 0.05. (**C**) ASMC migration was assessed by measuring the width of a “wound” in a confluent ASMC layer after 24 h. Bars represent mean ± SEM of independent experiments performed in ASMC of five asthma patients. The “control” depicts migration in the absence of IgE. * *p* < 0.05. (**D**) Representative immunofluorescence images of IgE-induced cytochrome c expression (green) in the presence and absence of signaling inhibitors after 24 h (600X magnification in enlarged boxes). Image analysis of 10 single cells in five individual ASMC lines (*n* = 50) is presented as the bar chart of the mean ± SEM. ** *p* < 0.01.

**Figure 6 ijms-20-00875-f006:**
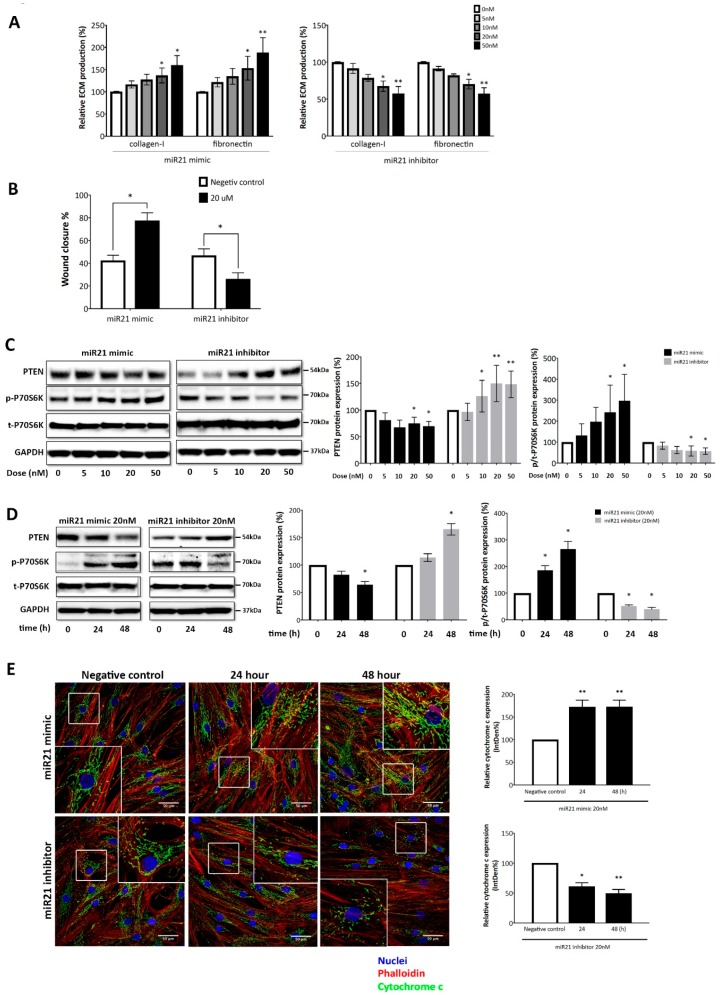
MicroRNA-21-5p is involved in ASMC remodeling via PTEN expression. (**A**) IgE-induced collagen-I and fibronectin deposition by ASMC was modified by overexpression of miR-21 mimics and miR-21 inhibitors. Bars represent concentration dependent (0–50 nM) deposition of collagen type-I and fibronectin as mean ± SEM. Experiments were performed as quadruplicates in ASMC of five asthma patients. * *p* < 0.05. ** *p* < 0.01. (**B**) IgE-induced ASMC migration was measured by the width of a “wound” after 24 h in the presence of either miR-21 mimics or miR-21 inhibitors (20 nM) in ASMC from five asthma patients. Bar charts represent the mean ± SEM. “control” indicates migration in the presence of control plasmids. Total wound width was set to 100%, * *p* < 0.05. (**C**,**D**) Representative Western-blot of kinetics for P70S6K phosphorylation and PTEN expression by overexpression of miR-21 mimics, or miR-21 inhibitors at various time points (0–48 h). Quantitative analysis of Western blots (*n* = 5) was performed by Image J and bars represent mean ± SEM. * *p* < 0.05. ** *p* < 0.01. (**E**) Representative immunofluorescence images for cytochrome c expression in the presence of miR-21 mimics (20 nM) or miR-21 inhibitors (20 nM). (600X magnification in enlarged boxes). Bar charts represent mean ± SEM of 10 single cells of five ASMC lines (*n* = 50). * *p* < 0.05. ** *p* < 0.01.

**Table 1 ijms-20-00875-t001:** Demographic information of tissue donors.

Samples	Age	Gender	Smoking	FEV1%
Control 1	68	Female	N.A.	101
Control 2	75	Male	N.A.	97
Control 3	81	Male	N.A.	89
Control 4	32	Female	0 pack/year	94
Control 5	65	Female	30 pack/year	89
Asthma 1	46	Female	10 pack/year	67
Asthma 2	79	Female	0 pack/year	102
Asthma 3	74	Female	20 pack/year	49
Asthma 4	52	Female	0 pack/year	92
Asthma 5	58	Male	10 pack/year	82

N.A. = not available.

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
