# Peer review of "IgE Downregulates PTEN through MicroRNA-21-5p and Stimulates Airway Smooth Muscle Cell Remodeling"

_ijms, 2019, doi:10.3390/ijms20040875_

Round 1

Reviewer 1 Report

In the present study Drs. Fang et al. investigated the remodelling effect and signalling pathways of IgE in the absence of allergens on isolated human primary ASMC from patients with allergic asthma compared to cells from control patients. The authors links IgE stimulated ASMC remodelling with up regulation of microRNA-21-5p.

The study makes interesting observations using an appropriate methodological approach. However, the results provided from the work are limited, the relevance of the conclusions is not so high as pretended and information about the performed experiments is missing.

In particular:

 - As the authors point on their conclusions, the described signalling pathways were independent of asthma and suggest microRNA-21-5p as target for ASMC remodelling. But no experiments were performed in the presence of allergens. By contrast, the Authors focussed full Introduction and Discussion on asthma and allergens. Thus the Authors should provide experiments showing the role of IgE and microRNA-21-5p in the presence of allergens. A second option would be to change Abstract, Introduction and Discussion pointing in the important results on signalling pathways and suggesting a role on the different diseased conditions.

- In the same regard, the Authors didn't study suppression of miR-21-5p as possible therapeutic target. What happens in the presence of allergens when miR-21-5p is inhibit?

- n numbers are not always clear. How many pictures/duplicates/experiments from how many patients. Furthermore, the study used 5 patients from each group but some figures show 4 or 6 patients in one group. 

- Many figures does not show control patients results.

- Where is the Graphic figure abstract?

- Format of References into the text is not always the same (i.e. reference 1 versus reference 2 or 12)

Author Response

Dear Reviewer 1,

We appreciate the your comment on our submission. Here we have submitted a revised version of our manuscript and prepared a point to point reply which is uploaded here as one attach file. We hope this will help us to address the concerned comments and provide sufficient answers. 

Thank you and best regards,

Michael Roth

Reviewer 2 Report

The manuscript written by Lei Fang et al is a very interesting scientific paper about a highly discussed topic about airway remodeling during allergic asthma. 

One major issue of this study is sample size (5 versus 5 ctrl) Is it a pilot study? More samples would be highly encouraged and could better support the main idea and results. Moreover, I missed a description about the patient age and gender. 

Another issue is some WB - I would prefer that authors present all the 5 ctrl and sample WB for the editor/reviewers as some figures are very subjective. 

Moreover, description of statistical analysis is very weak. Has to be improved. It is not clear which programs were used. 

Author Response

Dear Reviewer 2,

We thank you for the work on reviewing our submission to IJMS.

According to the editor and reviewer's suggestion, we have modified prepared a revised manuscript and point to point answer to your comment, which is attach as one word files here. We hope this can address properly to your comment and be sufficient to the revision. 

Thank you and best regards,

Michael Roth

Round 2

Reviewer 1 Report

Agree

Author Response

Dear Reviewer 1,

We thank you for the professional revision work and appreciate your comment and suggestion on our manuscript. 

Kind regards,

Michael Roth

Reviewer 2 Report

Authors made the significant changes to their manuscript, however, few changes have to be done. First of all, english language has to be revised again (e.g., line 54 - "allergen activated IgE activated pathway"- "activated" should be replaced with a synonym, line 57 "is involved ASMC" should be "is involved in ASMC" ir line 112 "due to by migration" should be changed to "due to migration", etc.).

I am still confused about SD in figure 1 C. SD in control samples is barely visible Is it so small?

Figure 4 A the right figure - lines showing significance are very confusing, I recommend to exclude line representing non-signficant results. Same figure C - one significance is lacking, please correct (Rap-IgE vs STat3-IgE).

Figure 5 C - right graph - some results are missing/not seen in the paper.   

Author Response

Dear Reviewer 2,

We thank you for the professional revision work, and appreciate your comment and suggestion on our manuscript, which helped us to improved the quality of our manuscript. We have prepared a point to point reply letter regarding your comment, as well as a trackable revised manuscript according to your suggestion. They are both uploaded for your further revision. 

Kind regards,

Michael Roth
